# Bioactive Compounds for Combating Oxidative Stress in Dermatology

**DOI:** 10.3390/ijms242417517

**Published:** 2023-12-15

**Authors:** Delia Turcov, Anca Zbranca-Toporas, Daniela Suteu

**Affiliations:** 1Faculty of Chemical Engineering and Environmental Protection “Cristofor Simionescu”, “Gheorghe Asachi” Technical University, 71 A Mangeron Blvd., 700500 Iasi, Romania; delia.turcov@gmail.com; 2Faculty of Medical Bioengineering, “Grigore T. Popa” University of Medicine and Pharmacy, Universitatii Street no. 16, 700115 Iasi, Romania; zbranca@roderma.ro

**Keywords:** antioxidant compounds, dermatology, natural compounds, oxidative stress, vegetal extracts

## Abstract

There are extensive studies that confirm the harmful and strong influence of oxidative stress on the skin. The body’s response to oxidative stress can vary depending on the type of reactive oxygen species (ROS) or reactive nitrogen species (RNS) and their metabolites, the duration of exposure to oxidative stress and the antioxidant capacity at each tissue level. Numerous skin diseases and pathologies are associated with the excessive production and accumulation of free radicals. title altered Both categories have advantages and disadvantages in terms of skin structures, tolerability, therapeutic performance, ease of application or formulation and economic efficiency. The effect of long-term treatment with antioxidants is evaluated through studies investigating their protective effect and the improvement of some phenomena caused by oxidative stress. This article summarizes the available information on the presence of compounds used in dermatology to combat oxidative stress in the skin. It aims to provide an overview of all the considerations for choosing an antioxidant agent, the topics for further research and the answers sought in order to optimize therapeutic performance.

## 1. Introduction

The formation of oxidants or pro-oxidants is a natural phenomenon within aerobic metabolism, but under certain conditions, which are increasingly common and persistent today, there is a persistent accumulation of reactive oxygen species (ROS), either by their overproduction or by exceeding the body’s antioxidant-defense capacity. Pro-oxidants are continuously produced by cellular respiration, other metabolic processes or by the action of external factors and are, in fact, highly reactive molecules. While oxidants in equilibrium act as molecular signals for the activation of the immune system, the continuous accumulation of reactive oxygen species (ROS) and reactive nitrogen species (RNS) with different structures that have a strong effect and exceed the capacity of local neutralization by antioxidants leads to multiple interactions with various other molecules and cells that damage proteins, membranes and genes.

This phenomenon increases in pathological conditions aggravating the underlying condition; this is oxidative stress.

Oxidative stress damages lipids, proteins, carbohydrates and DNA, leading to cell membrane damage, random fragmentation and cross-linking of their molecules, and even cell death induced by irreversible DNA fragmentation and lipid peroxidation.

These consequences are the molecular basis of cancer, neurodegenerative diseases, cardiovascular and autoimmune diseases, and diabetes [1]. These processes are accelerated by pre-existing pathological conditions.

The body’s response to oxidative stress can vary between exaggerated cell proliferation, cell-cycle disruption and arrest, apoptosis and necrosis depending on the type of ROS or RNS and their metabolites, the duration of exposure to oxidative stress and the antioxidant capacity at each tissue level.

Most discussions on oxidative stress refer to its general effects and induced changes in the body (on internal organs, blood vessels, eyes, the immune system, joints, the brain, etc.).

Some examples of the positive roles of free radicals are biochemical reactions such as hydroxylation, carboxylation, peroxidation and the reduction of ribonucleotides, with hydrogen peroxide (H_2_O_2_) being the best representative of these molecules as a secondary intra- and extracellular messenger that easily diffuses through biological membranes.

Several studies have demonstrated the interdependence between oxidative stress, the immune system and the inflammatory process, therefore, complete suppression of free radicals is not beneficial, having some risks [2].

In this context, the purpose of this article is to systematize the information regarding the existence of compounds used in combating oxidative stress at the skin level as applied in dermatology.

## 2. Current Considerations Regarding the Impact of Oxidative Stress on the Skin

The skin is the largest organ of the body and is of high structural and functional complexity. Therefore, new approaches to ensuring the quality of life and human health take into account all the elements that disrupt its proper functioning, including the effect of oxidative stress. The theory of oxidative stress and the increasing tendency to use natural extracts in this context have long been at the heart of many dermatocosmetic therapies, but these are far from exhaustive.

The most discussed is the general impairment caused by oxidative stress and the phenomena it triggers at different levels (internal organs, the circulatory system, the eyes, the immune system, the bone-joint system and the brain), but the skin is the main target of ROS during exposure to UV rays and pollutants. The skin is directly and constantly exposed to UV rays, which affect the skin structure. Moreover, the high content of molecular oxygen and the high concentration of polyunsaturated fatty acids contribute to the degradation of skin structures under unfavorable conditions.

UV rays are responsible for chronic skin damage, which can lead to photoaging and even malignant lesions [3]. In particular, UVB directly damages the structures of DNA, proteins and lipids. UVA is considered to have a wider impact through oxidative processes [4]. There are extensive studies that confirm a harmful and strong impact on the skin, including:(1)Promoting inflammatory processes;(2)Delaying and decreasing the efficiency of tissue regeneration and restoration processes;(3)Fragmentation and disorganization of collagen fibers;(4)Skin microbiome damage;(5)Damage to the transduction pathways of cellular signals;(6)Induction of apoptosis;(7)Modification of gene expression;(8)Changing the expression of integrin membrane receptors;(9)Participation of ROS in the initiation of degradation by metalloproteinases (enzymes that control the contact between fibroblasts and collagen fibers in the extracellular matrix—contact that ensures the integrity and functionality of the dermis) of type I collagen fibers by increasing the level of chronic exposure of these enzymes to UV and aging [5];(10)Impact on biomolecules containing thiol (-SH) groups affecting growth factor activity, inhibiting protein phosphatase, activating protein kinase and modifying transcription factors;(11)Modification of self-antigen proteins and the potential for initiating autoimmune conditions [2];(12)Protein lesions;(13)Peroxidation of membrane lipids, aggravating the destructive process;(14)Modification of calcium influx;(15)Increase in volume of mitochondria and their lization;(16)Formation of tumor cells, with a series of consequences developed in the form of the following diseases [5,6]: ● inflammation with erythema, oedema, local temperature increase and pain; ● premature aging of the skin; decreased local defense power; ● photoallergic reactions; ● autoimmune diseases; ● skin tumors [7].

Thus, there are skin diseases that are directly affected by oxidative stress and whose incidence has already increased alarmingly. The body has a complex antioxidant-defense system that is outpaced by the rate of formation and accumulation of ROS. This rate is increasingly accelerated by lifestyle and the unbeatable factors that lead to an increase in oxidative stress in the body: unprecedented technological development, urban lifestyles, high levels of pollution, damage to natural resources, an exhaustive pace of work and global impacts.

An important example of the primary role of signaling molecules exerted by ROS is the repair process following some injuries. Skin injuries are inevitable in the course of life, either due to internal or external causes. The healing process is a complex process that goes through three main phases: inflammation, proliferation and closure of the lesion.

Although ROS are involved in each of the three phases, they are mainly produced during the inflammatory phase to promote the healing process, protect from pathogens and regulate the angiogenesis process.

A test that examined the influence of H_2_O_2_ on wound healing was carried out and published in 2006. It demonstrated, for the first time, that small amounts of H_2_O_2_ are formed in vivo at the site of a wound, supporting and maintaining an angiogenesis mechanism that underlies the healing process. In addition, the study showed that the topical application of H_2_O_2_ at low concentrations, below 1%, on the wound site accelerated the healing process by maintaining angiogenesis, enhancing skin-stimulation mechanisms, increasing the affinity of growth factors for specific receptors and contributing to the remodeling of the extracellular matrix [8]. The level of H_2_O_2_ is higher in the early stages than in the final stages of the wound-healing process. The low level of ROS in the final stage of wound healing plays a mitogenic role by facilitating the entry of some cells into the proliferative phase of cell division at rest. During this stage, the role of antioxidants is crucial for maintaining optimal physiological healing and preventing aberrant and uncontrolled proliferation. This is proven by the presence, at the site of the lesion and in the final healing phase, of a high level of antioxidants (MnSOD, CuZnSOD, CAT and GPX) and the improper, deficient healing of lesions in the absence of them [9].

Another study suggests dividing dermatological conditions related to oxidative stress into two categories:
Those initiated by the ROS, in which free radicals are the basis of etiopathogenesis and aggravation:⮚Photoaging = premature aging of the skin manifested as wrinkles, laxity, spots and devitalization [10,11,12];⮚Atopic dermatitis = a chronic inflammatory condition manifested as dryness, itching and discomfort [12,13,14];⮚Psoriasis = a chronic autoimmune disease manifested by thick, squamous plaques, on large areas and causing major discomfort [15,16];⮚Acne = a condition characterized by the appearance of inflammatory or comedonian lesions on a hyperseborrhea background and damage to the pilo-sebaceous apparatus [17,18];⮚Rosacea = a chronic inflammatory condition characterized by redness, tenderness and fragility of the vascular walls in the face [19,20];⮚Melasma = the most difficult and extensive form of hyperpigmentation in the face [21];⮚Alopecia areata = partial or total hair loss [11];⮚Scleroderma = a rare and serious autoimmune condition manifesting as fibrosis and excessive thickening of the skin [11];⮚Vitiligo = an autoimmune manifestation characterized by the loss of skin pigment and discoloration [11];⮚Venous ulcer = serious injury to the calf due to venous insufficiency [11];⮚Vulgar pemphigus and follicle pemphigus = autoimmune conditions characterized by lesions similar to blisters or burns that are persistent and which do not give way to the usual therapies.Those in which ROS are produced as a result of the infiltration of polymorfonuclear leukocytes (PMNs) and macrophages into inflamed tissue:⮚Erythema = a change in the condition and appearance of the skin manifested as redness and inflammation [22];⮚Seborrheic dermatitis = an inflammatory condition in the scalp and face characterized by inflamed areas covered by squamous and accompanied by pruritus [23];⮚Allergic and irritating contact dermatitis = a pruriginous inflammatory condition triggered by exposure to certain allergens or irritants [11];⮚Lupus erythematosus = an autoimmune disorder affecting several organs in the body, including the skin [24];⮚Suppurative hydrosadenitis = painful inflammatory lesions that occur mainly in the areas of large folds (armpit, bikini and interfuse) [25].


The main mechanism of UV ray involvement in the pathogenesis of rosacea is the induction of reactive oxygen species followed by degradation of the extracellular matrix, favoring the development of the inflammatory process. Under UVB action, the synthesis of endothelial growth factor in keratinocytes increases, while UVA stimulates the degradation of dermal collagen. It also increases the amounts of pro-inflammatory mediators in the skin (IL1β, IL6, IL10, TNFα and CXCL8) [19,26,27,28]. It has also been found that the size of benign tumors is dependent on the proportion of DNA that undergoes oxidative damage, as identified by the oxidative damage marker of DNA, 8-hydroxy-2′-deoxyguanosin.

As far as skin aging is concerned, it is not only known with certainty that free radicals accelerate skin aging but also that the effects of oxidative stress in the skin are cumulative. The oxidation theory of skin aging is based on the hypothesis that the changes in aging skin are due to the accumulation of damage in the cellular macromolecules. The increased level of free radicals leads to cellular senescence, a physiological mechanism that prematurely terminates cell proliferation in response to damage that occurs during replication, the fundamental process in the cells of living organisms. These phenomena lead to the secretion of degradation enzymes that maintain or accelerate the phenomena of skin aging [29,30]. 

The relevance of biomarkers is justified in the confirmation of oxidative stress and, in general, the usual approach for the assessment of oxidative stress in biological systems involves the dosage of sensitive redox molecules as they increase or decrease the reaction time of the response to the phenomenon of oxidative stress. In general, oxidative stress markers have the following properties: they are chemically distinct and detectable, they increase or decrease during oxidative stress, they have a relatively long lifespan and they are not affected by other cellular processes (cell cycle, metabolism, etc.) [31].

Therefore, biomarkers of oxidative stress can be classified into one of four categories: oxidants, antioxidants, oxidation products and the measurement of the cellular redox balance (prooxidant/antioxidant ratio). Since chemical and biological systems communicate with each other, at least two of the above categories must be considered to confirm the presence of oxidative stress in tissues [32,33,34,35]. In the existing studies, there is no biomarker that has emerged as more relevant than others. The assessment of oxidative stress in the body is performed by measuring the concentration of the markers that characterize the prooxidant status.

As biomarkers for the generation of oxygen and nitrogen radical species, we mention myeloperoxidases (MPO), metalloenzymes Cu, ZnSOD and the dependence of GSHPx on SE. Among the most relevant biomarkers of changes induced by oxidative stress, there are the products of advanced glycation or glycooxidation processes (the reaction of the amino groups of lysine and arginine with the carbonyl groups of carbohydrates, resulting in hydroimidazolone, pentosidine, glucosepan and N-carboxymethyl-lysine), which are detected in serum and the skin. The peroxidation of polyunsaturated lipids leads to the formation of malondialdehide, the preferential marker of oxidative stress, which is easily detected in plasma. In turn, peroxidated lipids easily form radicals and singlet oxygen [36]. For antioxidant defense, o- thioredoxin (Trx), thioredoxin reductase (TrxR), NADPH and peroxydoxins (Prx) are important.

In-vivo tests of free radical presence and antioxidant efficiency are performed using fluorescence measurements, but these are limited by the short lifetime of some ROS or by access only to the epidermal corneal layer [37]. Other redox status indicators in vivo are the ratios of GSH/GSSG, NADPH/NADPSR, NADH/NADSR and TRX (thioredoxin) reduced/oxidized TRX. The progressive modification of redox status may represent the basic molecular mechanism in the processes of aging and the alteration of structural and functional cellular integrity [2].

Clinical and experimental assessments also use other methods to test oxidative stress and antioxidant status with specific indices:*OSI* = oxidative stress index; the ratio of oxidative stress to the antioxidant capacity of the body;*AOPP* (advanced oxidation protein products)/antioxidant capacity.

The concentration of oxidants and antioxidants in biological samples (plasma) can be calculated using the following parameters: TOS (total oxidative status), TOC (total oxidant capacity), TAS (total antioxidant status), TAC (total antioxidant capacity), the Oxy adsorbent test and the d-ROM test (the concentration of reactive oxygen metabolites) [12,37,38,39].

Studies show that antioxidant activity increases (CAT catalase and GSH-R glutathione reductase) and concentrations of α-tocopherol, ascorbic acid and glutathione are generally low in epidermis affected by intrinsic or extrinsic aging [40,41,42]. It has also been shown that a concentration gradient of antioxidants is high in the basal layer of the epidermis and low in the upper layers of the epidermis. This shows how important it is to supply the skin with natural or plant-derived antioxidants to ensure the protection of skin structures [37].

Therefore, antioxidants are among the most tested and used ingredients in pharmaceuticals, dermatocosmetics and dietary supplements. Another argument is that UVA and UVB photoprotectors act with minerals and physical filters on the skin surface by absorbing or reflecting certain wavelengths, being limited by the wavelength it can absorb, the lifetime of organic filters or other substances that can remove physical filters (dust, particles in the air, water and sweat). The protection provided by exogenous antioxidants is, therefore, supplementary to UV filters; they act in the deep layers of the skin, are present at the sites of UV-induced damage and neutralize free radicals or prevent their accumulation. In addition to anti-UV filters, which should be applied every 2 h, the exogenous antioxidants applied topically are stable and cannot be removed by washing or clothing and are effective for several days.

There are a number of clinical studies investigating the benefits of antioxidants, particularly for photoaging. The study protocols involve healthy subjects (men and women [43] or women only [44,45]), and the effect of long-term treatment with antioxidants is evaluated by studying their protective effects and the improvement in some of the oxidative stress-induced manifestations highlighted. Typically, subjects of different ages with fine or deep wrinkles, mild or moderate hyperpigmentation, dehydration and loss of skin luster, low tone and elasticity are included in such studies. The assessments are carried out at regular intervals after the start of the application of the topical antioxidant product investigated in the study. In this way, damage such as inflammation, sensitive areas, pigmentation, state of vascularization, amount and accumulation of sebum, hyperkeratosis (thickening of the skin), dehydration and skin texture are identified and highlighted.

The following compounds have been tested as antioxidants: idebenone, dL-alpha-tocopherol, kinetin, ubiquinone, L-ascorbic acid and dL-alpha-lipoic acid [4]. These, and other antioxidant molecules reported in the literature between 2005 and 2023, are summarized in Table 1, which highlights their nature, origin and biological effects [1,4,6].

The literature indicates a series of methods for evaluating antioxidant effects, such as [1,4,6]:-photochemilumimescence (the reaction to the lumen and light emission of radicals)—the concentration is expressed in nmol/L in vitro (i.e., in the case of Ibedeone);-the quantification of primary products (lipid hydroperoxide, high reactivity and cytotoxicity) in vitro (i.e., in the case of vitamin E);-the quantification of oxidation by-products (malondialdehyd) in vitro (i.e., in the case of kinetin);-the detection of sunburn cells (SBC) (idebenone provides a 38% reduction in SBC), antidimer timina antibodies in vivo (i.e., in the case of Ubiquinone).

There is a proven need for exogenous antioxidants, i.e., compounds that combat oxidative stress, prevent or stop the production of free radicals, contribute to their elimination and alleviate damage.

There are a variety of molecules that provide protection against oxidative stress and related diseases, making antioxidants one of the most studied and tested classes of pharmacological ingredients, with wide application in the pharmaceutical and dermocosmetic industries [39,46,47]. The selection of these ingredients is based on strict criteria defined by specific standards, medical-therapeutic guidelines and pharmacopoeias to ensure patient safety. The common interest of multidisciplinary teams is to reduce synthetic compounds and use natural compounds obtained through environmentally friendly extraction processes from tested natural sources to ensure the quality of the final product.

## 3. Bioactive Compounds for Combating Skin Oxidative Stress

Bioactive compounds with the potential to combat oxidative stress in the skin are presented systematically in Figure 1.

The biologically active compounds used in the treatment of oxidative stress in the skin can come from both natural sources and chemical synthesis processes. Both natural and synthetic compounds have advantages and disadvantages in terms of skin structures, tolerability, therapeutic performance, ease of processing or formulation and economic efficiency. Many synthetic molecules are derived or developed from natural compounds, so differentiation can be difficult. A schematic overview of these considerations can be found in Table 2.

Ultimately, the choice between synthetic and natural compounds depends on the situation and the needs of the skin. It is also worth mentioning the trend among patients and consumers who are currently opting for natural ingredients for reasons of skin safety but also for environmental reasons [48]. In terms of safety and therapeutic efficiency, the dermatologist is the authority, while in terms of production and processing efficiency, the principles of chemical engineering and economic calculation are decisive.

### 3.1. Bioactive Compounds as Antioxidant Ingredients in Dermatocosmetic Products for Combating Skin Oxidative Stress

An important category of substances for preventing and eliminating the effects of oxidative stress in the skin are bioactive substances with an antioxidant effect. These are compounds that counteract oxidative stress, prevent or stop the production of free radicals and, thus, eliminate and alleviate cell and tissue damage. Combating the harmful effects that can occur at the cellular level is based on a system that includes the following molecular strategies:prevention of damage;repair mechanisms that mitigate oxidative damage;physical protective mechanisms;antioxidant defense mechanisms.

In 2005, a protocol combining a series of five in vitro (biochemical and cell biological) and in vivo methods was developed for the first time to compare the protective capacity of the most common antioxidants used in photoprotective products. These combined in-vivo and in-vitro methods are the radical-scavenging capacity of photochemiluminescence, primary and secondary oxidation by-products (with the antioxidant ability to protect LDL and microsomal membranes), the UVB irradiation of keratinocytes (DNA damage in the cell culture) and the UVB irradiation of human skin to measure sunburn cell formation (SBC) damage [4].

In the absence of a standardized method for the characterization and comparative analysis of the properties and protective capacity against oxidative damage, the use of the Environmental Protection Factor (EPF) is proposed together with the SPF (Sun Protection Factor) and the IPF (Immune Protection Factor).

The literature points to the existence of numerous plant extracts and, thus, biologically active compounds that are of general benefit to skin health, with strong antioxidant activity, anti-inflammatory effects, protection against UV rays and photoaging, limiting effects on melanogenesis (the depigmenting effect, but also contribution to melanoma prevention), antitumor effects, and antiproliferative or antifungal effects. Some of the most promising classes in the dermatology industry are stilbenes (resveratrol), hydroxycinnamic acids (ferulic acid), carotenoids (crocin) and flavonoids (kaemferol), which have therapeutic potential as ingredients and are practically high for dermocosmetic formulations.

There are strong arguments that justify the development of a large number of studies on oxidative stress, a theory that was born sixty years ago, such as the acquisition of advanced information on the production and metabolism of reactive oxygen and nitrogen species, the identification of the biomarkers of oxidative damage, the demonstration of the direct link between oxidative stress and some acute and chronic diseases, and the search for the most effective natural antioxidants as bioactive molecules [49].

The dermatocosmetic industry is constantly evolving and has a direct impact on society. The need for new, effective products that are better tolerated by patients is leading research into advanced formulas that combine different active ingredients or similar ingredients in different concentrations. In addition, these ingredients can be obtained from domestic sources under more favorable economic conditions [46].

Topical treatment with antioxidants can safely contribute to the therapy of diseases triggered or aggravated by oxidative stress. The combination of different molecules with antioxidant activity in adapted doses is considered to be more effective. In addition, the associations of antioxidant molecules allow the administration of doses close to physiological ones, thus avoiding the pharmacological risk of toxicity or even the aggravation of the imbalance in the antioxidant system [47].

Standardized methods to compare the protective properties and capacity of antioxidants in vitro and in vivo, and to quantify their benefits in skin tissue, are constantly being developed. In addition, new effective solutions to combat and mitigate the harmful effects of oxidative stress at the cellular and tissue levels are constantly being sought and studied, using effective, safe and natural methods. Dermatology has the advantage of gently incorporating therapeutic ingredients into gestures that are already present in patients’ habits [50]. In this context, a number of studies have shown the importance of associating antioxidants to increase the protective activities in the cells and tissues of the body, with strong evidence for the prevention and alleviation of major diseases, whether systemic or cutaneous (Table 3). The additional benefits are related to the higher bioavailability and preservation of the active ingredients.

The topical application of antioxidants is intended to optimize skin protection and limit skin damage induced by ROS, including:The reduction of UVA-induced polymorphic light rash;The reduction of erythema induced in PUVA therapy (psoralen + UVA);Decreasing the production of sunburn cells (the cells that are massively and irreversibly affected by the action of UVB and which are subjected to an induced cell-death process by pro-apoptotic mediators, as a preventative measure against the formation of a malignant phenotype [4].

There are a number of long-established antioxidants, such as vitamin C, E, resveratrol and coenzyme Q10, that are used to combat and/or treat various dermatological conditions. Combating oxidative stress is a goal that is being considered by many professionals, and antioxidants play an extremely important role in this [39,46,56]. Other antioxidants are on the rise in dermatocosmetic formulas, including various polyphenols and even synthetic antioxidants such as idebenone [57,58].

Thus, when selecting the antioxidants to be administered, three essential properties have to be considered: good bioavailability, stability and selectivity for damaged cells. The formulation of dermatocosmetic products for daily care that contain antioxidant ingredients to protect skin structures is, therefore, an optimal prophylactic and therapeutic way to counteract oxidative stress and its consequences in the skin.

### 3.2. Antioxidants from Natural Sources

The bioactive compounds in natural extracts have played an important role in the pharmaceutical industry for decades, with dermocosmetics being one of the most important areas of application. The numerous benefits of natural bioactive ingredients for skin health have led to the development of a special product category, biocosmetics or cosmeceutics, as an intermediate form between medicine and cosmetics that contains biologically active ingredients and is considered similar to dermatological products with topical applications [59]. The following synonyms can be found in the literature: performing cosmetics, functional cosmetics, dermaceuticals, active cosmetics, phytocosmetics, skinceutics, nutricosmetics and dermocosmetics [60].

Natural active ingredients act as effective aids with important biorevitalizing, protective and trophic-supporting effects, which have significant pharmacological activity [61]. The types of natural substances contained in dermocosmetics are hormones, vitamins, enzymes, alkaloids, amino acids, antibiotics and antiseptics.

Plants are an important source of antioxidants, especially primary and secondary metabolites, which are continuously studied and utilized due to their pharmacological and ecological interest [62].

The main advantages of natural compounds are [39,46]:most of them are small, non-protein molecules with good penetrability;they can be obtained by extraction methods that do not compromise the quality of the ingredients concerned or the effect on the human body;they can also be obtained from spontaneous flora, without incurring production costs, or from plant residues left over after the extraction of other ingredients of interest using the modern zero-waste concept;they can be obtained by steam distillation from vegetable raw materials using organic or aqueous solvents and have a low molecular weight of less than 2000 Da (except biopolymers, natural gums, condensed tannins and some polysaccharides such as pectins and starches) [39,46].

In addition to improving the appearance of the skin due to their anti-aging action, biocosmetics or cosmeceutics have proven their effectiveness as active ingredients in the prevention or improvement of various dermatological diseases, exerting some effects simultaneously, such as antioxidant effects, anti-inflammatory effects, anticarcinogenic effects, cell proliferation modulation, angiogenesis, melanogenesis and protein synthesis [63,64].

## 4. Conclusions

The latest reports bring new details and evidence concerning the impact of oxidative stress in the skin. The discussion is so wide that little doubt remains regarding the involvement of free radical species in triggering and aggravating skin disorders and pathologies. A large variety of molecules offer protection from oxidative stress and related diseases, and among these, antioxidants have become the most studied and tested classes of pharmacological ingredients for the dermocosmetic industry. There is increasing evidence that antioxidant compounds counteract the damage induced by oxidative stress. Phenolic compounds, vitamins, carotenoids and minerals are the most investigated and effective molecules in combating the consequences of oxidative stress.

The major advantage of combining specific bioactive compounds can be the increase or the potentiation of the basic action, with better therapeutic results and greater satisfaction for patients and specialists, thereby improving the attributes of the product and also replacing harmful synthetic chemical compounds.

Modern extraction techniques, new associations between compounds, and the cultivation and study of plant resources with continuous exploitation highlight the sustainability and importance of natural products. Thus, established molecules are potentiated and continue to strengthen their place at the top of the list of dermatocosmetic ingredients alongside new promising natural compounds.

An important topic for further research is the discovery of the right synergisms and optimal associations between antioxidant compounds, as well as advanced research on the mechanisms and biological effects of new antioxidants gaining ground in dermatocosmetic formulas.

There are still several research directions that are looking for answers that can increase the therapeutic performance in many skin conditions, including:What safety measures can be taken in the selection of antioxidants for therapeutic purposes?What other combinations and doses of antioxidants increase therapeutic performance?Can all antioxidant molecules be evaluated comparatively?

Concerning dermatological evaluation, any clinical trial should take into account the complexity of redox phenomena, the flow of the oxygen radical species, and the subcellular location and type of cells.

## Figures and Tables

**Figure 1 ijms-24-17517-f001:**
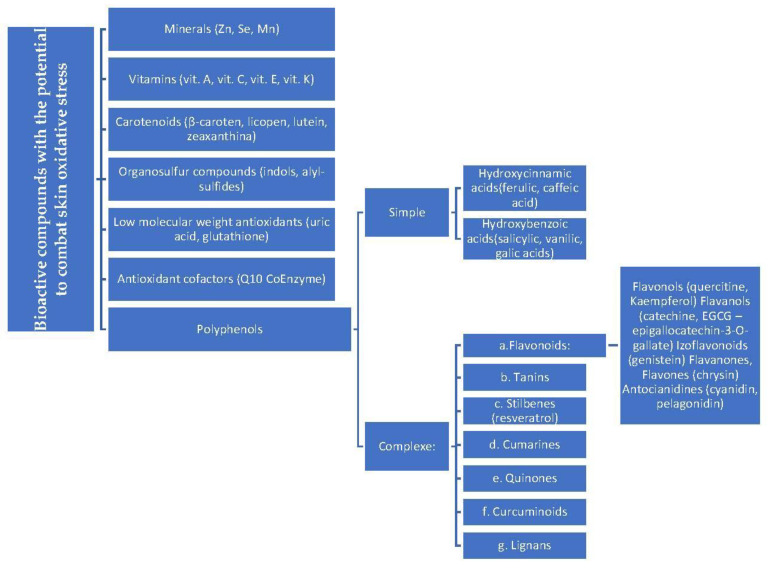
Several types of bioactive compounds with potential to combat skin oxidative stress.

**Table 1 ijms-24-17517-t001:** Antioxidant molecules reported in the literature between 2005 and 2023 as pharmacologically active, as prophylactic and therapeutic agents for conditions influenced by oxidative stress (aging, cancer, cardiovascular diseases, autoimmune, neurodegenerative diseases and diabetes).

Compound	Origin (Natural/Synthetic)	Type/Class	Antioxidant Mechanism of Action	Other Biological Effects
Idebenone	Synthetically	Analogue of ubiquinone (CoQ10)	It captures free radicals. It inhibits lipid peroxidation	It is considered to transfer electrons directly to complex III of the mitochondrial electronic transfer chain, restoring cellular energy generation (ATP)
Vitamin E	Natural—oily plants (rape, sunflower, soybean, corn, oil and seeds)	Vitamin—tocopherols and tocotrienols	Neutralization of singlet oxygen in the cell membrane. Prevention of lipid peroxidation (oxidation of unsaturated fatty acids such as arachidonic acid in the phospholipid membrane)	Cardioprotective, antitumor, prevents cataracts, neurodegenerative diseases and arthritis
Kinetin	Synthetic (from the natural pattern isolated from plants or originally from herring)	Cytokine (plant hormone)	Supports the endogenous antioxidant system	It modulates cell differentiation and division, and improves the barrier function of the skin
Ubiquinone (Coenzyme Q10)	All commercial sources of ubiquinone utilized in topical products are synthetically derived. For nutritional supplements, the fermentation processes of *Agrobacterium tumefaciens* are used	Polyphenols—quinone	The ability to transfer electrons; reduces ROS production; regenerates vitamin E; reduces DNA damage from keratinocytes and the production of UVA-induced metalloproteinases in fibroblasts; reduces mitochondrial oxidative damage	It supports cellular regeneration, tissue restoration, and elastin and collagen synthesis
Vitamin C	Natural—fruits and vegetables	Vitamins	Extensive removal of free radicals and regeneration of oxidized vitamin E	Cofactor involved in collagen synthesis. It inhibits melanogenesis; induces collagen synthesis; supports the production of skin-specific lipids; is neuroprotective
Lipoic acid	Generally synthetic	Organic compound with sulfur	Restoration of the endogen antioxidant system; regeneration of antioxidants (GSH, vitamin C, vitamin E and neutralization of SRO)	Essential cofactor for many enzymatic complexes (especially in aerobic metabolism, especially pyruvate dehydrogenase complex)
Resveratrol	Natural—*Vitis vinifera* sp., *Polygonum cuspidatum*	Polyphenols—stilbene	Induction of the antioxidant enzyme system. It inhibits damage and mutagenic action on DNA	Anti-inflammatory; antitumor, cell-cycle regulator; apoptotic; stimulates detoxification; antimicrobial; antiviral; antifungal. It inhibits the proliferation of keratinocytes
Lycopene	Natural—tomatoes	Carotenoid	It effectively removes free radicals and neutralizes singlet oxygen	Antitumor; prevents atherosclerosis and ophthalmological diseases
Lutein	Natural—vegetables	Carotenoid–xantophylls	Protects fibroblasts from UVA-induced oxidation. Prevents reduction of anti-dante catalase (CAT) and superoxide dismutase (SOD) enzymes. It is more stable under the action of oxidation than other carotenoids such as beta-carotene and lycopene	Anti-inflammatory; protects the eye tissues
Ferulic acid	Natural	Hydroxy-cinamic acids—polyphenolic compounds	It forms stable phenoxyl radicals	Antimicrobial; anti-inflammatory; antithrombotic; antitumor; vascular protector
Silymarin (silybinin up to 90%)	Natural—*Silybum marianum*	Flavonoid	It inhibits the production of SRO	Reduces UV-induced erythema and edema; hepatoprotective; regenerator; anti-inflammatory
Genisteine	Natural (soy) and synthetic	Flavonoid	It inhibits UVB-induced ROS generation	Anti-inflammatory; estrogen-mimetic; cell-cycle regulator
Pycnogenol (extract)	Natural (*Pinus pinaster* ssp. *Atlantica)*	Phenolic compounds (catechins, epicatechins and taxifolin); flavonoizi (procianidins/proantocianidins); phenolic acids (cinnamic acids and other glicosides)	Increases the synthesis of antioxidant enzymes; protects other antioxidants (vitamins C and E, and glutathione)	Reduces blood pressure. Increases the level of glucose in the blood. Relieves asthma and symptoms of allergic rhinitis. Improves lung function
Zeaxanthin	Natural—vegetables and fruits	Non-provitamin A carotenoids	Effectively removes free radicals; Neutralizes SRO; Prevents lipid peroxidation	Supports eye health; hepatoprotector
Rosmarinic acid	Natural	Phenolic compounds—pentacyclic triterpenes	Inhibits peroxidation of membrane lipids in situ	Anti-inflammatory; chemoprotector; neuroprotective
Ursolic acid	Natural	Phenolic—pentacyclic triterpenes	Reduces lipid peroxidation; it enhances the circulating antioxidants GSH, ascorbic acid and alpha-tocopherol	Anti-inflammatory; chemoprotector; antihyperlipidemic
Carnosic acid	Natural (*Rosmarinus officinalis*)	Phenolic—pentaciclic diterpene	Protects lipids (linolenic acid and monogalactosyldiacyl-glycerol) from singlet oxygen and hydroxyl radical	Anti-inflammatory; chemoprotector
Quercetin	Natural	Flavonoid	It regulates glutathione and its action; inactivates free radicals; donates a hydrogen atom; neutralizes the toxic effect of singlet oxygen by inactivating its ex-citation energy state; prevents lipid peroxidation	Prophylactic potential in osteoporosis, some types of tumors, and lung and cardiovascular conditions
Kaempferol	Natural	Flavonoid	It reduces the super-oxide anion, hydroxyl radical and peroxinitrite levels	Antitumoral; anti-inflammatory; antiproliferative
Phloretin	Natural (apple peel)	Dihydrochalcone polyphenol	It reduces the level of hydroxyl radon and prevents lipid peroxidation	Inhibits matrix metalloproteinases (MM P)-1 and elastase, enzymes that degrade connective tissue
Crocin	Natural—saffron (*Crocus sativus*)	Carotenoid	Reduces the level of several prooxidants; stimulates SOD and glutathione peroxidase activity (GPX)	Anti-inflammatory; immunomodulator; neuroprotective; antidepressant
Curcumin C	Natural	Curcumin	Induces glutathione-S-transferase. It inhibits the generation of free radicals and neutralizes them. It inhibits lipid peroxidation	Anti-inflammatory
Caffeic acid	Natural	Phenolic compounds—hydroxy-cinnamic acids	Relocation of unpaired electro-nits into the extended conjugated side chain	Anti-inflammatory; antitumorial; antibacterial; antifungal; prevents neurodegenerative diseases; prevents toxicity in chemotherapy
Nordihydro-guiaretic acid	Natural (unusual in topical or cosmetic products)	Polyphenols	Effectively removes SRO in vitro (peroxinitrite (ONOO^−^), oxygen singlet (^1^O_2_)•, hydroxyl radical (Oh), superoxide anion (O_2_^−•^), hydrogen peroxide (H_2_O_2_) and hypo-chlorous acid (HOCl))	Anti-inflammatory; anti-acne
Caffeine	Natural	Methylxanthin alkaloid	It inhibits the production of hydroxyl radical	Stimulator of the central nervous system; improves muscle contractility
Epigallo-catechin gallate	Natural (green tea)	Polyphenols	A large-scale effort to remove radicals. Inhibits the production of SRO and lipid pero-oxidation products. Protects the endogenous antioxidant system	Modulates the biochemical pathways involved in the inflammatory response, cell proliferation and the response of pro-tumor mediators.
Delphinidin	Natural—fruit	Flavonoids—anthocyanidins	It inhibits lipid peroxidation and the formation of 8-hydroxy-2′-deoxyguanosine (8-Ohdg)—a marker of oxidative stress on DNA and in carcinogenesis	Anti-inflammatory; antiproliferative
Niacinamide (vitamin B3)—nicotinic acid and nicotinamide	Predominantly synthetic	Water-soluble vitamin	It inhibits the generation of ROS; prevents the oxidation of lipids; proteins and DNA; improves the accumulation of intracellular calcium ions	Antioxidant—protects keratinocytes from oxidative stress
Polypodium leucotomos extract	Natural—fern *Phlebodium aureum*	Phenolic—p-cumarinic, ferulic, cafeic, vanilic, 3,4-dihidroxi benzoic, 4-hidroxi benzoic, 4-hidroxi cinnamic, 4-hidro-xicinnamoil-quinic and chlorogenic acids	It inhibits the generation of radicals under the action of UV, including superoxide anion. Reduces the production of reactive oxygen and nitrogen species	Photoprotective; immunomodulator; inhibits apoptosis by UV radiation; stimulates DNA repair
Squalene	Plant (olive oil, rice, corn and amaranth)	Triterpenoid	Oxygen-scavenging agent; quencer of singlet oxygen; prevents lipid peroxidation	Emollient; moisturizer; antitumoral

**Table 2 ijms-24-17517-t002:** Comparative presentation of the advantages and disadvantages of bioactive compounds that play a role in combating oxidative stress.

Natural Compounds		Synthetic Compounds	
Advantages	Disadvantages	Advantages	Disadvantages
- Better biocompatibility, being more easily metabolized into organisms - Better tolerability; non-toxic - Low molecular weight - Have multiple biological effects that act synergistically - Fewer long-term safety and environmental issues - More accessible, especially those from spontaneous flora - Cultivation and processing can have a positive impact on the sustainability and eco-nomy of a particular area or community	- Are less stable and have a lower period of validity, which can affect efficiency and tolerability - Can be difficult to produce in large quantities, becoming expensive and difficult to access - May vary in quality and efficiency with source, both from nature and processing methods (e.g., concentration varies according to source growth, harvesting and processing conditions) - Can trigger significant allergic reactions, especially in sensitive skin	- Can be designed to excert certain properties and functions unavailable from natural compounds - May have better stability and a longer period of viability - Can be produced in larger quantities, making them more affordable and cheaper - Can be formulated in association with another natural or synthetic co-posed to create a more powerful product - Some may be more effective at tagging certain skin problems (pigments-hard and acne)	- Possible adverse or allergic effects - Risk to long-term safety and environmental impact - Less biocompatibility in tissues - Are not yet fully regulated and do not have full scientific support for tolerability - The tendency in consumer behavior to avoid them for safety reasons

**Table 3 ijms-24-17517-t003:** Frequent associations between antioxidants in dermocosmetic products [14,39,51,52,53,54,55].

Antioxidant	Other Associated Antioxidants	Types of Dermocosmetic Products	Indications	Experimental Studies Evaluating Topical Use of Antioxidants (*)
Resveratrol	Vitamins C, E and A; baicalin; bisabolol; catechins; SOD, GSH; COENZYME Q10	- Day or night serum - Cream/emulsion - Lip balms - Facial masks - Gels - Exfoliation products - Peels - Sun-protection products - Skin lotions exposed to the sun - Concealer for localized spots or inflammatory lesions - Face/body oils - Soaps - Shampoos - Cleaning foams	- Skin laxity - Wrinkles, fine lines - Discromias, pigmentation spots (solar lentigo, ephelids, post-inflammatory hyperpigmentation and melasmas) - Dryness, dehydration - Redness, rosacea, acne - Mature, dull skin devoid of tone and brightness - Large pores, excess sebum, oxidative damage (photoaging) - Sunburn - Sensitive, irritated skin	- Tolerability - Antioxidant efficiency - Synergies between certain associations - Optimal concentration - Correlation between the concentration of the active ingredient and the roughness of the skin - Inhibition of certain key structures in the skin (hyaluronidase, an enzyme that degrades hyaluronic acid) - Optimal presentation forms and textures of the final product
Vitamin C	Vitamins E and A; resveratrol; ferulic acid			
Vitamin E	Vitamin C			
Coenzyme Q10	Vitamins C and E; Zn; Se			
Lycopene	Vitamins C and E; astaxanthin			
Ferulic acid	Vitamin C			
Niacinamide (vitamin B3)	Vitamins C and A			
Catechins (e.g., GEGC—epigallocatechin gallate from *Camellia sinensis* extract)	Zinc			

(*) Studies evaluating the topical use of antioxidants were not found for all antioxidant compounds in cosmetics and dermocosmetics.

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
