# Peer review of "Bioactive Compounds for Combating Oxidative Stress in Dermatology"

_ijms, 2023, doi:10.3390/ijms242417517_

Round 1
Reviewer 1 Report
Comments and Suggestions for Authors
Reviewer report paper number:
Bioactive compounds in combating oxidative stress in dermatology
The paper sound important as a review paper and has the potential to go forward to
the next step. The authors should revise the paper according to the following
comments:
- The abstract writes very generally and should be more focused on the main
aim of the review paper.
- Please add more references to the first paragraph of the introduction.
- The first paragraph of section 2 is not correct from biological point of view.
Especially the following paragraph: "the skin is the major target of ROS
during exposure to 75 UV and pollutants. The high level of molecular oxygen
and the high concentration of pol-76 yunsaturated fatty acids contribute to the
degradation of its structures under unfavorable 77 conditions. UVB directly
damages the structures of DNA, proteins, lipids."
Please correct all this section 2.
- I am not sure about the correctness of the number in the paragraph:
" These 187 phenomena lead to the secretion of degradation enzymes that
maintain or accelerate the 188 phenomena of skin aging " Please add a
relevant research reference.
- The data presented in paragraph: " The effect of long-term treatment with
antioxidants is evaluated through studies in-237 vestigating their protective
action and the improvement of some manifestations induced 238 by oxidative
stress. Typically, in such 3-6-month studies, women between the ages of 35
239 and 65 are included with fine or deep wrinkles, mild or moderate
hyperpigmentation, 240 dehydration, loss of skin brightness…. " is not
accurate and not based in the literature.
Reviewer 2 Report
Comments and Suggestions for Authors
The authors have put a great deal of work into creating this thorough and in-depth review of Bioactive compounds in combating oxidative stress in dermatology.
However, there are a few issues that must be made very clear to the reader before the article is considered for publication.
One of the main concerns is the English language. Namely, many compounds are not written correctly (e.g. Vitamina E and C, Quercitin, Polifenoli, Coenzima Q...). Then, some sentences are too long, causing confusion and/or unclear meaning.
Also, most of the tables are incomplete and/or incomprehensible.
Other comments are listed below:
Abstract
Lines 20-21: change “dermatological industry (stilbene (resveratrol)” into“dermatological industry are: stilbene (resveratrol)”
Lines 35-39: Too long a sentence that is hard to follow, please edit. I suggest to split the sentence.
​Lines 42-45: Please revise the sentence, so that it is comprehensible and has a logical consistency
Namely, oxidative damage of main macromolecules leads to membrane damage, DNA fragmentation, enzyme inhibition, and finally cell death. Remove enzymes, because enzymes are proteins. Fragmentation is mentioned two times, remove one of them.
Lines 47-48 – revise the sentence that you don't have the word „condition” twice
​Lines 54-55 change “…joints, etc). the brain).” in “…joints, brain, etc).”
Lines 73-79: This paragraph needs to be clearer. Point out why skin is prone to oxidative stress.
Lines 79-80: Who has a harmful and strong effect on the skin? Be precise, since is not clear.
Lines 107-112: The sentence is too long, it is difficult to follow. Please change it.
Lines 134-135: Add a reference.
Lines 190-218: Paragraphs regarding biomarkers need to be restructured.
Please indicate the most relevant biomarkers of oxidative stress (with type of samples) used in dermatological studies (not general) with supporting references.
In line with that, MPO is not a marker of oxidative stress removed in this paragraph. Also, ZnSOD and GSH-Px-Se are antioxidants, so these markers don't fit in the first part of the sentence.
It should not be separate indices such as TOS, TAC, d-ROM, etc. (lines 215-218) from the markers mentioned in the previous paragraph, if they are all relevant in dermatological studies (regard OS).
Lines 219-220: the sentence is not very clear. Do you mean - Studies show that antioxidant activities (CAT catalase, GSH-R glutathione reductase) intensify, while concentrations of α-tocopherol, ascorbic acid, and glutathione are generally low in the epidermis affected by intrinsic or extrinsic aging? Provide references for this.
Line 242: the word „décolleté” please correct.
Table 1 - is not complete. It is not clear why empty fields were left in the “test method” column, correct it.
Table 2 - I suggest replacing the word “they” with a hyphen (-) wherever it is mentioned in Table 2. For example “they are less stable" change to “ - less stable…”. Also, to make it easier to follow the text in the table, add a hyphen before each individual sentence.
Line 314-315: The sentence is not clear enough. What are the five methods? Cite references for it.
Line 326: see comment above (lines 20-21).
Line 331: „reactive oxygen and nitrogen species” replace with ROS and RNS
Table 3 - The table seems a bit confusing with half partitions. Does this mean that the antioxidants (resveratrol, vitamin C, E, Coenzyme Q...) have the same types of dermo-cosmetic products, indications etc.? If the answer is yes, adjust the table to make this clear. Also, remove Vitamin E; zinc, manganese, and selenium because they are repeated.
​Line 361: “..optimize antioxidant protection…” remove antioxidant, it is understood that it is protections based on its antioxidant action.
Lines 362-367: it is unclear whether it means the ways of action of antioxidants in order to protect the skin or not. In the first sentence (line 363) the reduction is stated in the others not. Please be precise.
Author contribution – is not filled.
Comments on the Quality of English Language
One of the main concerns is the English language. Namely, many compounds are not written correctly (e.g. Vitamina E and C, Quercitin, Polifenoli, Coenzima Q...). Then, some sentences are too long, causing confusion and/or unclear meaning.
Round 2
Reviewer 1 Report
Comments and Suggestions for Authors
The authors revised the paper based on my major comments
Author Response
Dear Reviewer #1
We thank you for your time spent for reviewing our revised manuscript and for your these last comments and suggestions that have been helpful to improve its quality.
Sincerely yours,
Prof.PhD.Eng. Daniela Suteu – corresponding author
Reviewer 2 Report
Comments and Suggestions for Authors
Thanks to the authors for the effort in rearranging the manuscript. The authors have satisfactorily addressed almost all my concerns. I have some minor considerations (see the comments below). After these are addressed, I would recommend the manuscript for publication in IJMS.
In Table 1 (Page 7) the word antifungal is mentioned 2 times (see Resveratrol - Other biological effects)
Lines 267-275: I suggest small changes/suggestions to make it clearer (see the bold words)
- quantification of primary products (highly reactive and cytotoxic lipid hydroperoxide) in vitro (i.e. case of vitamin E);
- quantification of oxidation by-products (malondialdehyde) in vitro (i.e. case of kinetin);
- detection of sunburn cell (SBC) (product of UVB damaging epidermal cells) and antithymine dimer antibodies (product of keratinocyte DNA damage) in vivo; (i.e. case of Ubiquinone).
Line 385: “..optimize antioxidant protection…” remove antioxidant, it is understood that it is protections based on its antioxidant action.
Comments on the Quality of English LanguageMinor editing of English language is required.
